# Assessing and investigating children with suspected bone and abdominal tumours: an e-Delphi consensus process

Dhurgshaarna Shanmugavadivel [1], Jo-Fen Liu [1], Ashley Gamble [2], Angela Polanco [2], Kavita Vedhara,[3] David Walker [1], Shalini Ojha [4,5]

[1]Lifespan and Population Health, University of Nottingham, Nottingham, UK
[2]Chief Executive's Office, Children's Cancer and Leukaemia Group, Leicester, UK
[3]Centre for Academic Primary Care, School of Medicine, University of Nottingham, Nottingham, UK
[4]Neonatal Unit, University Hospitals of Derby and Burton NHS Foundation Trust, Derby, UK
[5]Centre for Perinatal Research, Lifespan and Population Health, University of Nottingham, Nottingham, UK

**Correspondence to**
Dr Dhurgshaarna Shanmugavadivel;
shaarnashan@doctors.org.uk

## ABSTRACT

**Background** The incidence of childhood cancer has risen by 15% since the 1990s. Early diagnosis is key to optimising outcomes, however diagnostic delays are widely reported. Presenting symptoms are often non-specific causing a diagnostic dilemma for clinicians. This Delphi consensus process was conducted to develop a new clinical guideline for children and young people presenting with signs/symptoms suggestive of a bone or abdominal tumour.

**Methods** Invitation emails were sent to primary and secondary healthcare professionals to join the Delphi panel. 65 statements were derived from evidence review by a multidisciplinary team. Participants were asked to rank their level of agreement with each statement on a 9-point Likert scale (1=strongly disagree, 9=strongly agree), with responses ≥7 taken to indicate agreement. Statements not reaching consensus were rewritten and reissued in a subsequent round.

**Results** All statements achieved consensus after two rounds. 96/133 (72%) participants responded to round 1 (R1) and 69/96 (72%) completed round 2 (R2). 62/65 (94%) statements achieved consensus in R1 with 29/65 (47%) gaining more than 90% consensus. Three statements did not reach consensus scoring between 61% and 69%. All reached numerical consensus at the end of R2. Strong consensus was reached on best practice of conducting the consultation, acknowledging parental instinct and obtaining telephone advice from a paediatrician to decide the timing and place of review, rather than adult cancer urgent referral pathways. Dissensus in statements was due to unachievable targets within primary care and valid concerns over a potential overinvestigation of abdominal pain.

**Conclusions** This consensus process has consolidated statements that will be included in a new clinical guideline for suspected bone and abdominal tumours for use in both primary and secondary care. This evidence base will be translated into awareness tools for the public as part of the Child Cancer Smart national awareness campaign.

## WHAT IS ALREADY KNOWN ON THIS TOPIC

⇒ Childhood cancer is a global disease burden and is the leading illness cause of death in children between 1 and 14 in the UK.
⇒ Bone and abdominal tumours have some of the lowest survival estimates of childhood cancers.
⇒ They pose a diagnostic dilemma for clinicians, presenting with non-specific symptoms such as bone pain, limp and abdominal distension.

## WHAT THIS STUDY ADDS

⇒ Sixty-four new evidence-based statements on best practice, assessment, imaging and referral of children and young people presenting with key bone and abdominal symptoms.
⇒ Wide representation on the Delphi panel to include those from primary and secondary care resulting in pragmatic guidance to aid decision-making in both primary and secondary care.

## HOW THIS STUDY MIGHT AFFECT RESEARCH, PRACTICE OR POLICY

⇒ These data will form the basis of a new clinical guideline which will aid healthcare professionals to instigate investigation of those with signs and symptoms that could be due to a bone or abdominal tumour.
⇒ Empowering clinicians with this guidance will allow prompt recognition of those children with bone and abdominal tumours, improving their outcomes.
⇒ These data will be translated into public-facing materials for dissemination through a new awareness campaign called Child Cancer Smart to reduce the patient interval and promote early diagnosis.

## INTRODUCTION

Childhood cancer is often perceived to be rare; however, while individual childhood cancer types are rare, collectively they are more common than many believe. An individual's cumulative risk of cancer from birth to age 15 years is 1 in 450, with 1840 new cases diagnosed in children and young people (CYP) aged 0–15 each year in the UK.[1] Incidence of childhood cancer has increased by 15% since the 1990s. Although survival estimates are also increasing (from 77% in 2001 to 84% in 2016 across all childhood cancers[1]) progress in the UK lags behind other Western European countries.[1–4]

Many CYP experience delays in their cancer diagnosis.[5 6] Such delays are multifactorial.

Unlike in adult cancers, prevention and screening strategies are not available, therefore early diagnosis is key to optimising outcomes, reducing morbidity, mortality and treatment burden.[1] A possible explanation behind delayed diagnosis lies with childhood cancer symptomatology posing a diagnostic dilemma. Symptoms are often non-specific, mimicking other more common ailments such as gastroenteritis, migraines or can masquerade as pain attributed to minor injury. Furthermore, perceived rarity means childhood cancer is not often considered as a potential diagnosis by the parent or clinician. The 2015 National Institute for Health and Care Excellence (NICE) issued a guideline on 'Suspected cancer: recognition and referral'[7] which covers all ages. There is a real need for paediatric-specific guidance, as adult cancers manifest and present differently. This current guidance is directed at primary care with the 'end-point' being referral onto secondary care. Children with cancer experience diagnostic delay throughout the health service both at primary care and secondary care level and a referral from primary to secondary care can add significant length to the patient's diagnostic journey.

Furthermore, the recommendations lack a systematic evidence review and are based solely on expert consensus which notably did not include any paediatric oncologists or haematologists on the panel.

As a result, concern from the paediatric oncology community across the UK rose that the guidance was not fit for purpose.[8] A supplement to the NICE guideline was published in 2021 following a Delphi consensus process conducted among the Children's Cancer and Leukaemia Group (CCLG) membership.[9] A full systematic evidence review was not completed at this time due to the urgent need for expert child-specific guidance to be published.

Detailed tumour-specific diagnostic guidance as that produced for childhood brain tumours is needed in order to empower clinicians to make decisions about those who need investigation and accelerate referrals for CYP with high-suspicion symptoms promoting earliest possible diagnosis. Experience gained from developing and disseminating tumour-specific evidence-based guidance for diagnosis of childhood brain tumours through the HeadSmart campaign demonstrated that parents and clinicians were empowered by access to such guidance to identify the children who need investigation leading to a halving of the Total Diagnostic Interval (TDI) across the UK for children with brain tumours.[10] Using this methodology for other tumour types, grouped by location, could accelerate diagnosis and improve outcomes.

Prolonged diagnostic intervals have been widely reported in bone tumours. Of the BRIGHTLIGHT cohort in the UK, adolescents and young adults with bone tumours had the longest median symptom onset to diagnosis interval with multiple pre-referral consultations with general practice before diagnosis.[11]

Abdominal tumours involving sympathetic nervous system (neuroblastoma), renal/urogenital tracts (Wilms tumour, bladder tumours), liver (hepatoblastoma),

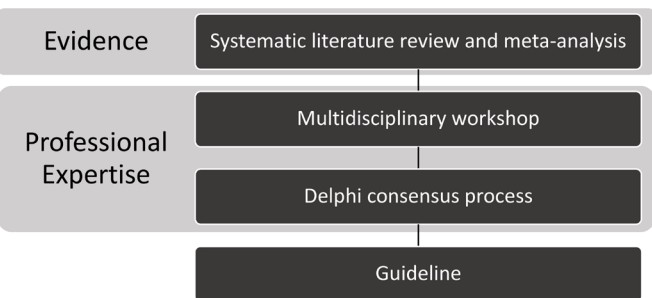

**Figure 1** Steps in guideline development.

lymphatic tissues (lymphoma/leukaemia) and connective tissues (soft tissue sarcomas) account for approximately 15% of all childhood cancers, representing a substantial proportion. Furthermore, there is strong evidence that children in the UK with a Wilms tumour have a significantly larger volume, a more advanced tumour stage at diagnosis and poorer survival compared with their European counterparts.[2 12]

Based on this collective evidence, we prioritised focusing on these as the next step to improving childhood cancer awareness.

As the questions of specificity, referral pathways, investigation indications and acceptable waiting times for CYP with suspected cancer cannot easily be addressed by quantitative research methods alone, a Delphi consensus process[13 14] was employed to use professional expertise to incorporate the evidence from systematic reviews into a clinical guideline.

## METHODS
### The Delphi consensus process
The Delphi consensus process was conducted as the final step of the gold-standard Appraisals of Guidelines Research and Evaluation (AGREE II) approach to clinical guideline development (figure 1).[15]

A Guideline Development Group (GDG) consisting of seven members with experience in developing clinical guidelines (two paediatric oncologists, a representative member of the Royal College of Paediatrics and Child Health (RCPCH), the chief executive of the CCLG, a general practitioner (GP), a paediatric registrar, research assistant and parent representative) oversaw the guideline development. The initial stage comprised a systematic review and meta-analysis of the literature. This provided contemporary evidence regarding the signs and symptoms of bone and abdominal tumours in CYP.[16 17]

### Multidisciplinary workshop
A multidisciplinary workshop was held on 12 November 2019 with 15 participants (seven male, eight female). Participants consisted of doctors from primary, secondary and tertiary care (two GPs, three paediatric emergency department consultants, three paediatricians, three paediatric oncologists and two paediatric surgeons), and two parents of CYP who have been diagnosed with cancer.

The systematic review identified 29 bone tumour symptoms/signs and 97 abdominal tumour signs/symptoms but only those that occurred in 2% or more of the patients were reported as a list of pooled proportions. This list was reviewed and refined by the multidisciplinary workshop participants to four bone tumour symptoms (pain, swelling, mass/lump and restricted movement or limp) and four abdominal tumour symptoms (mass, haematuria, pain and distension). For each sign/symptom, the group were asked to consider a set of questions on clinical presentation, assessment and investigation strategies and suggest answers or possible approaches to care. The questions for discussion were:

► How would this sign/symptom present?
► How should a healthcare professional (HCP) assess a child/young person presenting with this sign/symptom?
► How should an HCP determine whether this presenting sign/symptom could be due to a bone or abdominal tumour (specificity)?
► What factors influence the specificity of this sign/symptom?
► What are appropriate thresholds for referral and/or investigation for a child/young person presenting with this sign/symptom?
► What would you regard as best practice for referral and/or investigation of a child/young person presenting with this sign/symptom?

The discussions were recorded, and contemporaneous notes taken. The discussion points and notes were collated and translated into a series of statements by the GDG at the end of the workshop. These were sent back to the workshop participants to check accuracy and content. The workshop participants did not participate in the Delphi consensus survey.

## Delphi consensus surveys

The statements derived from the workshop were sent to HCPs using a modified e-Delphi consensus process.[14] Invitation emails were sent out to HCPs to join the Delphi panel; it included doctors from primary, secondary and tertiary care across a wide range of specialties who may encounter CYP with these symptoms in their daily practice (general paediatricians, GPs, community paediatricians, paediatric surgeons, paediatric radiologists, paediatric orthopaedic surgeons, paediatric oncologists/haematologists and emergency paediatricians). HCPs were recruited from the CCLG membership, Association of Paediatric Emergency Medicine and through general practice mailing lists.

The survey was built using Jisc Online Surveys and the Delphi panel members were asked to rank their agreement with the statements by means of a 9-point Likert scale (1=strongly disagree; 5=neither agree nor disagree; 9=strongly agree) with the ability to comment by free text.

## Definition of consensus

Based on existing guidelines, statements were defined as reaching consensus if 70% or more of the Delphi panel respondents rated the statement as either 7, 8 or 9.[8] Statements were rejected if 25% or less of the Delphi panel respondents rated the statement 7, 8 or 9.[13]

The rankings for each statement were collated. Any statement achieving the predetermined level of consensus was accepted. Statements not reaching consensus were rewritten and reviewed by the multidisciplinary workshop participants. Free-text comments were used to help structure the rewritten statement prior to being reissued in round 2 (R2).

## Patient and public involvement

Two parent representatives with experience of childhood cancer diagnosis volunteered to participate in the multidisciplinary workshop and helped revise the statements following feedback from the Delphi panel.

## Ethical approvals

This Delphi consensus process is part of clinical guideline development and HCPs were recruited for their expertise by virtue of their professional role. Ethical approvals were not required; however, consent was obtained from all participants prior to the workshops and Delphi process, with explicit consent being asked for recording of sessions and use of quotes or feedback as part of the process.

## RESULTS

One hundred and fifty HCPs practising in the UK were invited to take part. One hundred and thirty-three agreed to take part consisting of 57 GPs, 28 general paediatricians, 18 paediatric emergency consultants, 13 paediatric surgeons, 6 paediatric radiologists and 11 paediatric oncologists. Sixty-five statements were derived from the workshop and reviewed by the RCPCH guideline development team as part of the endorsement process. The statements were split into categories: best practice in conducting the consultation (referral, imaging, predisposing factors), bone tumours (general, bone pain, swelling, mass/lump, restricted movement/limp) and abdominal tumours (general, abdominal pain, abdominal mass, haematuria, abdominal distension).

The first round of the Delphi consensus process was open from 2 March to 23 March 2020. A Delphi survey containing 65 statements was sent out to all 133 participants. During this period, the COVID-19 pandemic was declared and the round was paused. The survey was reissued on 9 September 2020, allowing those who still wished to complete it to do so. In total, 96 (72%) participants completed round 1 (R1). The second round was open between 9 November and 30 November 2020 and was completed by 69 of 96 (72%) respondents who had taken part in R1. All 65 statements reached numerical consensus after two rounds (figure 2).

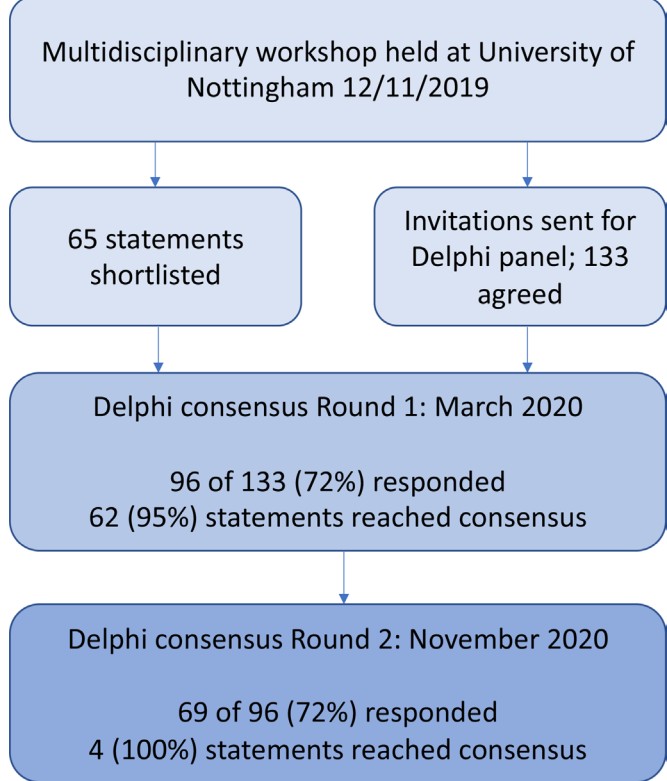

**Figure 2** The Delphi consensus process.

## Round 1

Ninety-six of 133 (72%) participants responded to the first round.

Sixty-two of 65 (95%) statements achieved numerical consensus. Three of 65 (5%) statements did not achieve numerical consensus and no statements were rejected.

Of the 62 consensus-reaching statements, 29 (47%) gained more than 90% consensus, 25 (40%) statements gained between 80% and 90% consensus and 8 (13%) statements between 70% and 80% consensus.

Despite achieving numerical consensus, statement 3, which read 'if a child or young person (CYP) warrants another appointment for the same problem, the timing of this review should comply with national diagnosis of all cancers (currently, diagnosis or all clear should be given to the patient within 4 weeks)', was revised and included in R2 due to comments outlining ambiguity in the wording.

Statements 8, 19 and 53 did not reach consensus, with 69%, 66% and 61% rating the statements 7–9, respectively. These were revised by the multidisciplinary workshop participants based on the comments received and sent out in R2 (table 1).

## Round 2

Sixty-nine of 96 (72%) participants responded to the second round.

All four statements reached numerical consensus at the end of R2. However, the statement regarding the investigation of abdominal pain which was revised and accepted was deemed similar to another statement and was omitted from the final list.

The final 64 statements are presented in figures 3–5.

**Table 1** Statements that did not reach consensus

| Original statement for R1 | Common themes in comments | Revised statement for R2 |
|---|---|---|
| **Statement 3**<br>**R1 consensus 72%**<br>If a child or young person (CYP) warrants another appointment for the same problem, the timing of this review should comply with national diagnosis of all cancers (currently, diagnosis or all clear should be given to the patient within 4 weeks). | **Comments**<br>▶ Only relevant if there is a concern around cancer.<br>▶ Depends on the clinical situation. | **R2 consensus 89.8%**<br>Any healthcare professional deciding to review a patient to diagnose or exclude cancer should ensure that the timing of the review does not exceed the national 4-week limit to access a diagnostic test and obtain the result. |
| **Statement 8**<br>**R1 consensus 69.1%**<br>Discuss concerns with a secondary healthcare professional before referring a CYP from primary care in which the differential diagnosis includes a possible tumour (low index of suspicion) to ensure the CYP is seen within 2 weeks. | **Comments**<br>▶ Would use 2-week wait.<br>▶ No available telephone service for paediatrics.<br>▶ Ambiguity over who to call. | **R2 consensus 89.6%**<br>Discuss concerns over the telephone with the consultant paediatrician hotline or local equivalent service before referring a CYP from primary care in which differential diagnosis includes a possible tumour to ensure the CYP is seen within the most suitable time frame (ideally within 2 weeks). |
| **Statement 19**<br>**R1 consensus 66%**<br>Measure and plot height and weight measurements for CYP with signs or symptoms suggestive of a bone tumour on age-appropriate growth charts. | **Comments**<br>▶ Not feasible in a 10 min consultation within primary care.<br>▶ Weight is more important than height.<br>▶ Growth charts not easily accessible in primary care. | **R2 consensus 88.36%**<br>Be aware that weight loss can be a sign of a bone or abdominal tumour. Measure weight and compare to any previous measurements in CYP with signs or symptoms suggestive of a bone or abdominal tumour. Plot these measurements on age-appropriate growth charts if available to you to monitor change when reviewing symptoms. |
| **Statement 53**<br>**Round 1 consensus 60.9%**<br>Request ultrasound imaging of the abdomen and kidneys for persistent (occurring on most days for a 2-week period) abdominal pain of unknown cause. | **Comments**<br>▶ Abdominal pain is a very common symptom especially in primary care.<br>▶ Would not request ultrasound if not red flags. | **R2 consensus 77.9%**<br>Request ultrasound imaging of the abdomen and pelvis for new persistent abdominal pain (occurring on most days for a 2-week period) of unknown cause where there is another symptom present from the checklist. |

R1, round 1; R2, round 2.

The final statements after two rounds of a Delphi consensus process
## Section A Best Practice (Statements 1-13)

| | | | |
|---|---|---|---|
| General | 1 | Explicitly ask young people, parents and carers about their concerns regarding what the cause of symptoms are in any consultation. | 99.0% |
| General | 2 | If a parent/carer expresses concerns about a bone/abdominal tumour this should be reviewed carefully. If a tumour is unlikely, explain why and give appropriate safety netting advice. | 72.0% |
| General | 3 | Any healthcare professional deciding to review a patient to diagnose or exclude cancer should ensure that the timing of the review does not exceed the national 4-week limit to access a diagnostic test and obtain the result. | 89.8% |
| General | 4 | Offer a telephone or in person interpreter if the patient, parent / carer or healthcare professional are not fluent in English or Welsh. | 95.9% |
| General | 5 | Be aware that low parental educational level, social deprivation and lack of familiarity with the UK healthcare system may be associated with diagnostic delay. Consider a multi-disciplinary approach for these families (for example health visitor liaison) and provide clear written safety netting for when to seek further medical advice. | 90.7% |
| General | 6 | Be aware that the presence of complex neuro-disabilities or other communication difficulties (eg, Autistic Spectrum Disorder) may also be associated with diagnostic delay. Care should be taken to elicit concerns from parents or carers that know them best. | 98.0% |
| Referral | 7 | In primary care, discuss concerns with your local consultant paediatrician hotline or the paediatric consultant on call the same day if there is a high index of suspicion regarding a possible bone or abdominal tumour in a CYP. | 90.1% |
| Referral | 8 | Discuss concerns over the telephone with the consultant paediatrician hotline or local equivalent service before referring a CYP from primary care in which differential diagnosis includes a possible tumour to ensure the CYP is seen within the most suitable timeframe (ideally within 2 weeks). | 89.6% |
| Referral | 9 | Abdominal tumours can progress rapidly over days, increasing in size and causing pressure effects (eg, breathing difficulties or bowel obstruction and ischaemia). If there is suspicion of an abdominal tumour this should be discussed in person with a paediatrician to decide the most appropriate time for review, which will often be the same day. | 92.7% |
| Imaging | 10 | Request a plain x-ray in anteroposterior (AP) and lateral view as the initial investigation for a CYP who has a suspected bone tumour. This should not delay referral from primary care. | 76.2% |
| Imaging | 11 | Request an ultrasound of the abdomen and pelvis as the initial investigation for a CYP who has a suspected abdominal tumour. This should not delay referral from primary care. | 80.4% |
| Predisposing factors | 12 | Be aware that some predisposing factors are associated with an increased risk of childhood bone tumours. Verify the presence of predisposing factors with parents/patients as they may lower the threshold for referral and investigation. | 79.4% |
| Predisposing factors | 13 | Be aware that some predisposing factors are associated with an increased risk of childhood abdominal tumours. Verify the presence of predisposing factors with parents/patients as they may lower the threshold for referral and investigation. | 83.5% |

**Figure 3** Percentage consensus for final best practice statements. CYP, children and young people.

## DISCUSSION
### Summary
These data provide professionally agreed backbone for best practice for use in a new clinical guideline in the assessment and investigation of CYP with suspected bone or abdominal tumours. When developed, this will be the first clinical guideline to be published specifically for these tumour types, and the second stand-alone guidance for CYP following the HeadSmart guidance for childhood brain tumours.[18]

The initial round achieved a consensus in 95% of the statements sent out for review and all statements reached consensus within two rounds. This was higher than expected, a testament to the work of the multidisciplinary workshop participants in clarifying and negotiating statements that were applicable across specialties. Those statements that did not reach consensus achieved between 60.9% and 69.1% and required minor revisions before achieving consensus in R2.

### Best practice in general approach
Overall, there was strong consensus regarding the best practice of conducting the consultation, taking into account parental concern or instinct which has been shown to be an important factor in diagnosis of childhood cancer.[19 20] Referral pathways for childhood cancers include an urgent referral process for suspected cancer but has been subject to criticism. Studies have shown that only 2% of childhood cancer is actually diagnosed via this route and the distress caused to parents being referred using this pathway far outweighs the rate of pick-up.[21 22] Furthermore, for those for whom cancer is the diagnosis, this pathway could potentially add a further 2-week delay, depending on waiting times for appointments, before

The final statements after two rounds of a Delphi consensus process
## Section B Bone Tumours (Statements 14-38)

| | | | | |
|---|---|---|---|---|
| **General** | 14 | Take a detailed history, including the presence or absence of the other symptoms on the list above, history of injury, predisposing factors and a family history for CYP presenting with symptoms suggestive of a bone tumour. | 99% |
| | 15 | Be aware that bone tumours can present with systemic symptoms such as fever, malaise and weight loss. Ask about these associated symptoms when seeing patients with other symptoms suggestive of a bone tumour. | 96.8% |
| | 16 | Be aware that bone tumours causing spinal cord compression can affect bladder, bowel dysfunction. Ask for the presence of urinary or faecal incontinence (and erectile dysfunction in adolescent males) when taking a history, especially if the presenting complaint is back pain. | 94.8% |
| | 17 | If there is an associated injury, take a detailed history of the injury including the mechanism of injury and timings of the onset of symptoms after the injury occurred. | 95.9% |
| | 18 | Examine the limb or joint in question, the joint above and below, and perform a neurological and musculoskeletal examination (eg, paediatric Gait, Arms, Legs and Spine assessment, pGALS) in a CYP with signs/symptoms suggestive of a bone tumour. | 90.7% |
| | 19 | Be aware that weight loss can be a sign of a bone tumour. Measure weight and compare to any previous measurements in CYP with signs or symptoms suggestive of a bone or abdominal tumour. Plot these measurements on age-appropriate growth charts if available to you to monitor change when reviewing symptoms. | 88.3% |
| | 20 | Be aware that an initial normal x-ray does not exclude a bone tumour. If symptoms or clinical suspicion persists, a referral to secondary care is warranted. In secondary care, a discussion with a radiologist about the most appropriate repeat imaging is advised. | 88.6% |
| | 21 | Be aware that pelvic bone tumours may not initially show on an x-ray. If the persistent symptom is pelvic pain and the x-ray has been reported as normal, referral to secondary care is warranted. In secondary care, discuss with a paediatric radiologist for further advice on imaging. | 85.6% |
| **Bone pain** | 22 | Ask about the presence of the other symptoms of a bone tumour (swelling, palpable lump, restricted movement/limp, fever, weight loss, back pain and bowel/bladder/erectile dysfunction) in a CYP presenting with persistent bone pain (occurring on most days for a 2-week period). | 99% |
| | 23 | Be aware that children aged younger than 4 years, or those with communication difficulties, are frequently unable to describe pain; their behaviour eg, withdrawal, holding their leg, not weight bearing may indicate bone pain. Look for these signs on examination. | 98% |
| | 24 | Be aware that an initial normal x-ray does not exclude a bone tumour. If symptoms or clinical suspicion persists, consider discussion with a paediatric radiologist and repeat x-ray or further imaging. | 85.6% |
| | 25 | Request x-ray imaging for persistent bone pain (occurring on most days for a 2-week period). In primary care, request of imaging should not delay referral to secondary care. | 83.5% |
| | 26 | Request x-ray imaging for localised bone pain that is waking a child or young person at night. | 85.5% |
| | 27 | Request x-ray imaging for unexplained bone pain (ie, without any preceding injury). | 81.5% |
| | 28 | Be aware that x-ray imaging is not always the most suitable imaging modality for persistent bony back pain. Discuss with a paediatric radiologist to decide upon the most suitable imaging of choice. | 87.6% |
| **Swelling** | 29 | Ask about the presence of the other symptoms of a bone tumour (bone pain, palpable lump, restricted movement/limp, fever, weight loss, back pain and bowel/bladder/erectile dysfunction) in a CYP presenting with persistent swelling (occurring on most days for a 2-week period). | 96.9% |
| | 30 | Be aware that delayed diagnosis has been associated with attributing a red warm swelling to infection despite no improvement with antibiotics. Arrange to see the CYP at the end of the course of antibiotics to assess response. If there has been no response, consider discussion with secondary care for advice or referral. | 87.7% |
| | 31 | Request x-ray imaging for persistent swelling (present for 2 weeks or more) rapidly increasing in size. | 75.2% |
| | 32 | Request x-ray imaging for persistent swelling (present for 2 weeks or more) not resolving despite treatment with regular anti-inflammatories OR antibiotics. | 78.3% |
| **Bone mass/ lump** | 33 | A bony mass or lump which is increasing in size can be a sign of an underlying bone tumour and requires referral to secondary care. | 95.9% |
| | 34 | Ask and examine for the other signs and symptoms suggestive of a bone tumour (bone pain, swelling, limp/restricted movement, fever, weight loss, back pain and bladder/bowel/erectile dysfunction) in CYP with a lump/mass. | 93.8% |
| | 35 | Request x-ray imaging for a rapidly increasing bony lump or mass. This should not delay referral from primary care. | 86.6% |
| **Restricted movement/limp** | 36 | Ask about the presence of the other symptoms of a bone tumour (bone pain, palpable lump, restricted movement/limp, fever, weight loss, back pain and bowel/bladder/erectile dysfunction) in a CYP presenting with restricted movement or limp. | 93.9% |
| | 37 | Have a high level of concern for a CYP who is normally highly active or sporty but is no longer able to play sport due to the presenting symptom. | 87.6% |
| | 38 | Request x-ray imaging for a CYP who is non-weight bearing or has restricted movement despite adequate analgesia. | 83.5% |

**Figure 4** Percentage consensus for final bone tumour statements. CYP, children and young people.

The final statements after two rounds of a Delphi consensus process

## Section C Abdominal Tumours (Statements 39-63)

| | | | |
|---|---|---|---|
| General | 39 | Take a detailed history including the presence of any of the other symptoms on the list above, any predisposing factors and a family history. | 94.8% |
| | 40 | Examine the abdomen, including external genitalia and hernia orifices, visualise and palpate the spine and perform a neurological examination in a CYP with signs/symptoms that may be due to an abdominal tumour. | 93.6% |
| | 41 | Perform a urine dipstick to exclude infection and identify any protein or blood that would warrant referral to secondary care. | 89.7% |
| | 42 | Consider recording blood pressure if the correct size cuff is available to identify hypertension. | 74.2% |
| | 43 | Be aware that weight loss can be a sign of an abdominal tumour. Measure weight and compare to any previous measurements in CYP with signs or symptoms suggestive of a bone or abdominal tumour. Plot these measurements on age-appropriate growth charts if available to you to monitor change when reviewing symptoms. | 88.3% |
| Abdominal mass | 44 | Ask about the presence of the other symptoms of an abdominal tumour (abdominal pain, haematuria, abdominal distension, weight loss, fever, malaise, jaundice, bone pain, neurological symptoms and bowel/bladder/erectile dysfunction) in a CYP presenting with an abdominal mass. | 99% |
| | 45 | Be aware that abdominal masses can cause neurological symptoms due to pressure on the spinal cord. The pressure can cause children to present as "off legs" or refusal to weight bear. Examine the abdomen when a CYP presents with refusal to weight bear. | 94.9% |
| | 46 | Be aware that diagnostic delay is associated with failure to perform an abdominal examination in a child who is distressed/crying. Offer to examine the child once they have settled or ask a colleague to perform the examination. | 94.8% |
| | 47 | Request ultrasound imaging of the abdomen and pelvis for a CYP with a palpable abdominal mass (unless this is felt to be faeces, in which case a review of the CYP after disimpaction is important). | 83.6% |
| | 48 | Request ultrasound imaging of the abdomen and pelvis for a CYP with suspected hepatomegaly or splenomegaly on examination. | 83.5% |
| Abdominal pain | 49 | Ask about the presence of the other symptoms of an abdominal tumour (haematuria, abdominal distension, weight loss, fever, malaise, jaundice, bone pain, neurological symptoms and bowel/bladder/erectile dysfunction) in a CYP presenting with persistent abdominal pain. | 92.8% |
| | 50 | Examine the abdomen in a CYP with abdominal pain to elicit any masses or hepatomegaly and/or splenomegaly. | 99% |
| | 51 | Be aware that if the tumour is retroperitoneal, the presenting complaint may be back pain. Examine the abdomen in a CYP presenting with back pain. | 95.9% |
| | 52 | Be aware that delayed diagnosis has been associated with attributing abdominal pain to constipation despite no improvement with laxatives. Assess response to laxatives by reviewing the CYP at regular intervals and taking a full history and examining their abdomen. | 87.7% |
| | 53 | Request ultrasound imaging for persistent abdominal pain with one or more other symptoms that may be due to an abdominal tumour (abdominal distension, mass, haematuria, weight loss, fever, malaise, jaundice, bone pain, neurological symptoms and bowel/bladder/erectile dysfunction) | 86.6% |
| Haematuria | 54 | Ask about the presence of the other symptoms of an abdominal tumour (abdominal pain, mass, weight loss, fever, malaise, jaundice, bone pain, neurological symptoms and bowel/bladder/erectile dysfunction) in a CYP presenting with haematuria in the absence of another known cause. | 96.9% |
| | 55 | Do a careful abdominal examination to elicit any potential retroperitoneal mass in all CYP with haematuria. | 92.7% |
| | 56 | Be aware that delayed diagnosis has been associated with assuming persistent haematuria is due to a UTI despite no response to antibiotics. Arrange to see the CYP again at the end of the course of antibiotics to assess the response. If there is no response, consider discussion with paediatrician for advice. | 82.4% |
| | 57 | Request ultrasound imaging for unexplained persistent (occurring for 2 weeks or more) macroscopic haematuria. | 87.7% |
| | 58 | Request ultrasound imaging of the abdomen and pelvis for haematuria plus one or more other symptoms that may be due to an abdominal tumour (abdominal pain, abdominal distension, weight loss, fever, malaise, jaundice, bone pain, neurological symptoms and bowel/bladder/erectile dysfunction). | 87.6% |
| Abdominal distension | 59 | Ask about the presence of the other symptoms of an abdominal tumour (abdominal pain, haematuria, weight loss, fever, malaise, jaundice, bone pain, neurological symptoms and bowel/bladder/erectile dysfunction) in a CYP presenting with an abdominal distension. | 97.9% |
| | 60 | Be aware that abdominal distension caused by a tumour will not fluctuate but will increase in size progressively over time. Arrange another appointment to review the symptoms and re-examine the CYP. | 83.5% |
| | 61 | Be aware that delayed diagnosis has been associated with attributing abdominal distension to constipation despite no improvement with laxative treatment. Assess response to treatment in these CYP by seeing them at regular intervals. | 84.5% |
| | 62 | Be aware that delayed diagnosis has been associated with failure to examine the abdomen at review of a CYP on treatment for constipation. Palpate the abdomen in CYP who are being reviewed for constipation. | 94.9% |
| | 63 | Request ultrasound imaging of the abdomen and pelvis for a CYP with rapidly increasing abdominal distension. This should not delay referral from primary care. | 83.5% |

**Figure 5** Percentage consensus for final abdominal tumour statements. CYP, children and young people; UTI, urinary tract infection.

any investigation is requested. Reaching clear consensus on obtaining telephone advice from a paediatrician to decide the timing and place of review will ensure that all CYP are appropriately triaged based on their history. This is in line with standards set by the RCPCH in the 2015 'Facing the Future' publication stating that GPs assessing or treating children should have access to immediate telephone advice from a consultant paediatrician.[23] Despite this guidance, there were some comments from primary care stating a lack of availability of advice in their region.

One recurring theme among the comments was the regional differences within primary care. A number of GPs highlighted that there was no ability to request paediatric imaging directly from primary care and if there was, this could take weeks or even months to take place. A caveat was added to the investigation statements to state that requesting an investigation should not delay a referral.

### Dissensus in R1

The dissensus in three statements was due to unachievable targets within primary care and valid concerns over a potential overinvestigation of a common symptom, respectively.

Measuring and plotting height and weight takes place routinely within secondary care paediatrics. However, GPs raised concerns around the ability to complete this within a 10 min appointment and the lack of availability of appropriate growth charts, either electronically or in paper format. They also queried the importance of this over other information available in the history and examination that would warrant referral regardless of the child's height. Based on this feedback, the GDG felt that weight was important and while a single measurement would not be useful, weight loss is an important feature to elicit, especially in the context of reattendance for non-specific symptoms. This highlighted the importance of ensuring guidance is feasible for both primary and secondary care.

Investigating abdominal pain also caused much debate due to the frequency of children seen with ongoing abdominal pain for whom constipation or functional abdominal pain is the most likely diagnosis. Primary care sees many more children with these symptoms that do not enter secondary care. It was agreed that abdominal pain should not be seen as a single symptom and investigation was warranted if there were other persistent symptoms present.

### Strengths and limitations

The methodology has followed the AGREE II tool[15] which is the gold-standard process for guideline development. The multidisciplinary team and Delphi panel comprised a wide range of clinicians from both primary and secondary care. Representation of professional groups who see these symptoms in CYP on a daily basis has ensured that the final statements are pragmatic for all.

The workshop also included parent representatives ensuring that the guideline also meets the needs of the children, young people and their families.

The attrition of participants between rounds gave a smaller cohort than expected. This was largely due to the pandemic as the first round was open during the time the UK went into lockdown and many panel members were front-line workers in emergency and primary care. However, the majority of consensus was gained in the first round prior to further attrition and as part of the RCPCH endorsement process, there will be further stakeholder review of the guideline prior to publication.

### Implications for practice

Childhood cancer poses a diagnostic dilemma for clinicians due to non-specificity of symptoms. Earlier diagnosis offers clinical presentations with smaller and less advanced tumours, requiring less therapy and therefore better outcomes. This guideline, which will be published by summer 2023, will empower clinicians to investigate CYP for a prompt diagnosis but to also identify those CYP who do not need any further investigations.

These data will also undergo further development methodologies to allow translation into decision support tools and awareness materials. These will be disseminated through public messaging, raising awareness of the signs and symptoms of abdominal and bone tumours through a new national awareness campaign called Child Cancer Smart in September 2023.

### CONCLUSION

This consensus process has provided expert guidance that will be included in a new clinical guideline for suspected bone and abdominal tumours for use in both primary and secondary care. This evidence base will be translated into awareness tools for the public as part of the Child Cancer Smart campaign.

**Acknowledgements** We would like to thank all those who participated in the multidisciplinary workshop to devise the initial set of statements and review and revise the results of the consensus rounds. We would also like to thank all the clinicians who participated in the Delphi consensus process during the pandemic, many of whom were redeployed to front-line care. We would like to acknowledge and thank the work of the RCPCH evidence team for their advice and support with this consensus process.

**Contributors** DS, J-FL, SO and DW conceived the guideline development process. DS, J-FL, DW, AP, AG, SO and KV were involved with the design of the process. DS drafted the manuscript which was reviewed and edited by all. DS is the guarantor for this work.

**Funding** This study was funded through a Doctoral Research Fellowship (DRF-2018-11-ST2-055) awarded to DS by the National Institute of Health Research (NIHR). The systematic reviews and meta-analyses that informed this guideline development were funded by Cancer Research UK through the Population Research Committee Early Diagnosis Innovation Grant (C59357/A22874) awarded to DS.

**Competing interests** There are no competing interests to declare.

**Patient and public involvement** Patients and/or the public were involved in the design, or conduct, or reporting, or dissemination plans of this research. Refer to the Methods section for further details.

**Patient consent for publication**  Not applicable.

**Provenance and peer review**  Not commissioned; externally peer reviewed.

**Data availability statement**  Data are available upon reasonable request. All data are available upon reasonable request.

**ORCID iDs**

Dhurgshaarna Shanmugavadivel http://orcid.org/0000-0002-1912-4543
Jo-Fen Liu http://orcid.org/0000-0001-5796-7878
Ashley Gamble http://orcid.org/0000-0002-0708-0918
Angela Polanco http://orcid.org/0000-0002-4619-0773
David Walker http://orcid.org/0000-0002-1753-0839
Shalini Ojha http://orcid.org/0000-0001-5668-4227

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
