## [Reviewer comments · BMJ Paediatrics Open]

ARTICLE DETAILS

TITLE (PROVISIONAL)	Assessing and investigating children with suspected bone and abdominal tumours: an e-Delphi consensus process.
AUTHORS	Shanmugavadivel, Dhurgshaarna Liu, Jo-Fen Gamble, Ashley Polanco, Angela Vedhara, Kavita Walker, David Ojha, Shalini

VERSION 1 – REVIEW

REVIEWER	Reviewer name: Dr. Kathy Pritchard-Jones Institution and Country: UCL Great Ormond Street Institute of Child Health, United Kingdom of Great Britain and Northern Ireland Competing interests: None
REVIEW RETURNED	08-Dec-2022

GENERAL COMMENTS	This is an important and interesting piece of work from a leading group in promoting earlier diagnosis of childhood cancer. They have applied principles developed and tested as part of the HeadSmart programme to promote early diagnosis of brain tumours in children and young people (CYP). In this manuscript, they describe work focussing on presenting symptoms and recommended initial investigation of suspected bone and abdominal tumours. However, other childhood cancer types may present with similar symptoms (eg bone pain in leukaemias, abdominal swelling in lymphomas, weight loss and persistent fevers in both). The authors should provide a rationale for why they chose to focus on two anatomical locations for CYP cancers rather than considering all non-brain CYP cancers in this guideline development process. Abstract Page 3, line 40 – should 5-point Likert scale be 9-point Likert scale? Page 3, line 23 – Children present with symptoms not with a suspected tumour. Please rephrase. page 4, line 9/10 – the term ‘two week wait’ is meaningless to a non-UK audience and should be rephrased or explained. Section: What is already known on this topic. Page 5, line 11 – I doubt that childhood cancer is the leading illness cause of death in all children aged over 1 yr in every country in the world. Suggest that the word ‘globally’ is rephrased. Introduction Page 7, line 12 – The opening statement “Childhood cancer is not rare” could be misleading to the intended general audience. Suggest expanding it to clarify that whilst individual childhood cancers are rare, collectively they are not.
--

Page 8, line 28 – the sentence “Currently, UK survival estimates for bone and abdominal tumours in CYP are the lowest of all childhood cancers” should be qualified – as there are several abdominal tumour types that have high survival rates (Wilms tumour, germ cell tumours).

Methods are clearly described

Results section and figures 3-5 (final statements)

Some statements are highly duplicative e.g numbers 20 and 24 under bone tumours; 10 & 25. What is the planned process to consolidate these in developing the clinical guideline?

Statement 47: the use of the term ‘felt to be’ in this statement is ambiguous, at least to this reviewer. Was this not questioned in the Delphi process? “Request US imaging..... for a CYP with a palpable abdominal mass (unless this is felt to be faeces” – the use of the word ‘felt’ could be taken to mean that faeces was palpated per rectum rather than the more likely interpretation that it is the belief of the examiner that the mass is composed of faeces (i.e. only assumed to be faeces). Suggest rephrasing.

Statement 57: a single episode of macroscopic haematuria in a child is generally considered sufficient for urgent investigation and referral. How was the requirement for this symptom to be persistent over a 2 week time period decided upon in order to warrant concern of cancer? Did this come from the literature review or the expert opinion of the HCPs in the workshop?

Discussion, page 15, lines 46-50. The following sentence should be rewritten. “Referral pathways for childhood cancers currently use the Two Week Wait pathway but has been subject to criticism”. Whilst there is a so-called ‘two week wait’ pathway provided by NHS England for children’s cancers, it is rarely used in practice, as stated in the next sentence. Also the terminology of ‘two week wait’ is unique to the British NHS and should be rephrased for an international readership. Suggest to replace ‘currently use’ with ‘include an urgent referral process for suspected cancer’ or something similar.

The following sentence on lines 57-60, same page, also needs rephrasing: “this pathway adds a further two week delay before any investigation is requested”. The whole point of the two week wait urgent referral pathway is that it should take no more than 14 days for the child to be seen, and not that everyone ‘waits’ two weeks!

Page 16, lines 51-53. More detail should be provided about why some GPs commented on “the lack of availability of appropriate growth charts”. This seem an extraordinary omission that primary health care staff assessing children should not have easy access to growth charts either in paper format or on their practice computers. Is the lack of access due to lack of time to do the measurement or difficulty in locating the appropriate growth chart, either physically or electronically?

Page 17, lines 40-43. The statement: “The workshop also included parent representatives ensuring that the guideline also meets the needs of the children, young people and their families” is listed as a strength, but how can the authors be certain of the veracity of this statement when only two parents were included in the workshop, and no young people or cancer survivors themselves? Did they do any broader validation with a larger number of family representatives? Is this planned for the next stage of the work?

It would be helpful for the reader if intended timelines for the

	production of the intended clinical guideline and public awareness tools could be included at the end of the discussion. Reference 1 needs to be rewritten – England PH means Public health England, CTYA should be spelt out in full and a URL and accession date should be given for accessing and downloading the report. Similar concerns for refs 6 and 19.
--	--

REVIEWER	Reviewer name: Dr. Bob Phillips Institution and Country: University of York, United Kingdom of Great Britain and Northern Ireland Competing interests: None
REVIEW RETURNED	29-Nov-2022

GENERAL COMMENTS	Comments to the Author This is the report of a modified, COVID-interrupted, Delphi survey to produce a series of statements related to the investigation of suspected bone and abdominal tumours in children and young people. It draws from two prior systematic reviews of symptoms in these groups of conditions, and follows a similar method to the 'HeadSmart' approach undertaken previously. I have reviewed a previous iteration of this paper and this is greatly improved. Two Minimal Points for Consideration: Methods/MDWorkshop: "The systematic review identified five bone tumour symptoms" - were these the only symptoms associated or were they selected through good diagnostic accuracy (fair sensitivity and specificity)? Please can this be explained? Other, and overly geekily; the submitting author needs to fiddle with their EndNote library formatting to make refs 1,5,9 and 19 work properly in the author field.
---

REVIEWER	Reviewer name: Dr. Juliet Gray Institution and Country: 27 Locks Road, United Kingdom of Great Britain and Northern Ireland Competing interests: None
REVIEW RETURNED	04-Dec-2022

GENERAL COMMENTS	This is a very useful consensus statement, which addresses and important area. The Delphi process used to achieve the consensus statement was robust and is well described. Minor comments for amendment: i) In abstract be consistent about including absolute numbers and percentages ii) I am not sure I agree that abdominal tumours have the worst prognosis of all children's cancer (line 28, pg 8) - Wilms tumours and Burkitt's lymphoma have an excellent prognosis - and the outcome for bone/abdominal tumours is generally better than many CNS tumours. iii) It may be helpful to cite published data demonstrating that Wilms tumours diagnosed in UK have a higher staging than in other European countries.
---

VERSION 1 – AUTHOR RESPONSE

We would like to say thank you to all 3 reviewers for taking the time to review our manuscript and are grateful for the expertise and comments to enhance the readability and accuracy of this piece of work. Our response to each of the comments are detailed below.

Response to Reviewer 1: Dr. Bob Phillips, University of York, Leeds Childrens Hospital

1. Methods/MDWorkshop: "The systematic review identified five bone tumour symptoms" - were these the only symptoms associated or were they selected through good diagnostic accuracy (fair sensitivity and specificity)? Please can this be explained?

Apologies, this had been lost in translation due to word count limitations. We have reworded to explain this in the manuscript.

The initial symptom lists were as reported in each of the studies, using pooled proportions of those symptoms that occurred in 2% or more of the patients. These symptom lists for bone and abdominal tumour presentations were taken to the expert multidisciplinary workshop where some were grouped together, others moved to associated symptoms (eg fever) and some removed for duplication (eg pain and swelling).

For example, for bone tumours, the search strategy identified 15477 papers. 713 papers were reviewed in full; 11 met the inclusion criteria, describing the symptoms/signs at diagnosis in 1246 children. 29 symptoms/signs were recorded but only those that occurred in 2% or more of patients are reported. These were pain (76%), swelling (21%), fever (4%), history of trauma (3%), functional limitation (3%), palpable mass (3%), pain and swelling (2%), volume increase (2%), limp (2%) and pathological fracture (2%).

In this example, our final list was bone pain, swelling, mass/lump (which included volume increase) and restricted movement or limp (again grouped together). Fever will be listed as an associated symptom in the guideline. History of trauma was not thought of as a presenting symptom but has been included as a diagnostic pitfall.

2. Other, and overly geekily; the submitting author needs to fiddle with their EndNote library formatting to make refs 1,5,9 and 19 work properly in the author field.

This has been amended and should now work correctly.

Response to Reviewer 2: Dr. Juliet Gray

Minor comments for amendment:

i) In abstract be consistent about including absolute numbers and percentages

Thank you, this has been corrected.

ii) I am not sure I agree that abdominal tumours have the worst prognosis of all children's cancer (line 28, pg 8) - Wilms tumours and Burkitt's lymphoma have an excellent prognosis - and the outcome for bone/abdominal tumours is generally better than many CNS tumours.

Thank you for highlighting this, we have removed this statement and instead clarified the reason for focussing on bone and abdominal tumours.

iii) It may be helpful to cite published data demonstrating that Wilms tumours diagnosed in UK have a higher staging than in other European countries.

Thank you, this was an omission on our part, we have added the citation of the published data in addition to the editorial.

Response to Reviewer 3: Dr. Kathy Pritchard-Jones, UCL

1. The authors should provide a rationale for why they chose to focus on two anatomical locations for CYP cancers rather than considering all non-brain CYP cancers in this guideline development process.

Thank you this is an important point. The rationale has been added to the introduction on page 8.

2. Abstract

Page 3, line 40 – should 5-point Likert scale be 9-point Likert scale?

Page 3, line 23 – Children present with symptoms not with a suspected tumour. Please rephrase.

Page 4, line 9/10 – the term 'two week wait' is meaningless to a non-UK audience and should be rephrased or explained.

Thank you, we have amended the Likert scale and rephrased as per the suggestions.

3. Section: What is already known on this topic.

Page 5, line 11 – I doubt that childhood cancer is the leading illness cause of death in all children aged over 1 yr in every country in the world. Suggest that the word 'globally' is rephrased.

Apologies, we have rephrased this.

4. Introduction

Page 7, line 12 – The opening statement "Childhood cancer is not rare" could be misleading to the intended general audience. Suggest expanding it to clarify that whilst individual childhood cancers are rare, collectively they are not.

Thank you, it is indeed something we have spoken about frequently! We have expanded this as per your suggestion.

Page 8, line 28 – the sentence “Currently, UK survival estimates for bone and abdominal tumours in CYP are the lowest of all childhood cancers” should be qualified – as there are several abdominal tumour types that have high survival rates (Wilms tumour, germ cell tumours).

We have removed this statement and expanded on the rationale for choosing abdominal and bone tumours as the focus of this guideline, further down on page 8.

5. Results section and figures 3-5 (final statements)

Some statements are highly duplicative e.g numbers 20 and 24 under bone tumours; 10 & 25. What is the planned process to consolidate these in developing the clinical guideline?

Yes, these comments are duplicative. The reason for this is because they fall under different headings of the guideline. For example, number 20 falls under ‘general recommendations for bone tumours’ heading, whereas number 24 falls under the specific symptom of bone pain. This was a deliberate distinction for those looking solely at the presenting symptom for guidance as opposed to the general section and reflects the same model used for the HeadSmart guidance.

Statement 47: the use of the term ‘felt to be’ in this statement is ambiguous, at least to this reviewer. Was this not questioned in the Delphi process? “Request US imaging..... for a CYP with a palpable abdominal mass (unless this is felt to be faeces” – the use of the word ‘felt’ could be taken to mean that faeces was palpated per rectum rather than the more likely interpretation that it is the belief of the examiner that the mass is composed of faeces (i.e. only assumed to be faeces). Suggest rephrasing.

Thank you for highlighting this and we can understand how this is ambiguous. In the Delphi consensus process, there were many free text comments about ambiguity or wording, even for those statements which reached consensus and many statements were amended based on this at the time. Interestingly, this particular ambiguity did not come up in the comment section and so we cannot change the wording for this manuscript as they form part of the results of the consensus process. However, we will put this forward for consideration of amendment to the multi-disciplinary workshop group for when the full guideline is reviewed, using these expert review comments as an audit trail.

Statement 57: a single episode of macroscopic haematuria in a child is generally considered sufficient for urgent investigation and referral. How was the requirement for this symptom to be persistent over a 2 week time period decided upon in order to warrant

concern of cancer? Did this come from the literature review or the expert opinion of the HCPs in the workshop?

This was from the expert opinion. It was agreed that the majority of children with frank macroscopic haematuria would present to their GP or to A&E promptly and be investigated immediately. The general practitioners felt that they see infection as a more common cause of microscopic and less frank macroscopic haematuria and so would treat with antibiotics initially without arranging an ultrasound. For these cases, we stipulated that if there was no improvement at 2 weeks and an ultrasound had not been done, it should be requested.

6. Discussion

page 15, lines 46-50. The following sentence should be rewritten. “Referral pathways for childhood cancers currently use the Two Week Wait pathway but has been subject to criticism”. Whilst there is a so-called ‘two week wait’ pathway provided by NHS England for children’s cancers, it is rarely used in practice, as stated in the next sentence. Also the terminology of ‘two week wait’ is unique to the British NHS and should be rephrased for an international readership. Suggest to replace ‘currently use’ with ‘include an urgent referral process for suspected cancer’ or something similar.

This is an important clarification for international readership and has been amended as suggested.

The following sentence on lines 57-60, same page, also needs rephrasing: “this pathway adds a further two week delay before any investigation is requested”. The whole point of the two week wait urgent referral pathway is that it should take no more than 14 days for the child to be seen, and not that everyone ‘waits’ two weeks!

We have rephrased to highlight the pathway as a potential source of delay so this should now be clearer.

Page 16, lines 51-53. More detail should be provided about why some GPs commented on “the lack of availability of appropriate growth charts”. This seem an extraordinary omission that primary health care staff assessing children should not have easy access to growth charts either in paper format or on their practice computers. Is the lack of access due to lack of time to do the measurement or difficulty in locating the appropriate growth chart, either physically or electronically?

Working in secondary care where all clinic patients have their height, weight and head circumference routinely measured before coming into the clinic room, this certainly was surprising to us. However, GPs highlighted that in secondary care we have nursing staff available in clinic to do this for us which is not the case within general practice. What we also learnt is that there is great variation from practice to practice, not just about availability of growth charts but also the availability of investigations and telephone advice. Certainly, time is a factor, and they could not fathom doing this for each child they see within 10 minutes without nursing help. Depending on which IT system they use, the growth chart may or may not be available on their practice computers. For those who did

not have them on practice computers, it seemed that paper charts were not frequently available to them. But most importantly, they questioned the importance of this information over other red flags that would be elicited in the history and examination for cancer that would be more likely to prompt them to refer in.

Page 17, lines 40-43. The statement: “The workshop also included parent representatives ensuring that the guideline also meets the needs of the children, young people and their families” is listed as a strength, but how can the authors be certain of the veracity of this statement when only two parents were included in the workshop, and no young people or cancer survivors themselves? Did they do any broader validation with a larger number of family representatives? Is this planned for the next stage of the work?

We have followed the RCPCH endorsement process which currently does not include PPI in the development process but does involve them as part of the stakeholder review process once the guideline is drafted. We wanted to involve parent representatives in the workshop to ensure that we had input from those who have had experience which was highly valuable, in particular to highlight to the experts about the importance of gut instinct. We will ask children and young people to review the quick reference guide through our local Young Persons Advisory Group (YPAG). We will also consult RCPCH&Us with the draft guideline as a stakeholder through the RCPCH endorsement process. The translation of the guideline into awareness tools will involve cancer survivors, young people and parents as we feel this will be where their insight will be valuable for public awareness.

It would be helpful for the reader if intended timelines for the production of the intended clinical guideline and public awareness tools could be included at the end of the discussion.

Thank you, we have added this to the manuscript. The clinical guideline and accompanying awareness tools will be published within the next 6 months (pending endorsements) with the public awareness tools launched during Childhood Cancer Awareness Month in September.

Reference 1 needs to be rewritten – England PH means Public health England, CTYA should be spelt out in full and a URL and accession date should be given for accessing and downloading the report.

Similar concerns for refs 6 and 19.

Apologies, these have been amended accordingly.

VERSION 2 – REVIEW

REVIEWER	Reviewer name: Dr. Kathy Pritchard-Jones
-----------------	--

	Institution and Country: UCL Great Ormond Street Institute of Child Health, United Kingdom of Great Britain and Northern Ireland Competing interests: None
REVIEW RETURNED	02-Feb-2023

GENERAL COMMENTS	The authors have addressed my comments to my satisfaction. I suggest in the editing process that the term 'generic' adult cancer pathways (line 12, page 4, clean version) is substituted by 'urgent referral' to be more accurate (as there are several types of generic adult cancer referral pathways described). I remain unhappy about the statement that macroscopic haematuria in a very young child is not a reason for immediate concern, but accept the authors' explanation that this was the 'anecdotal' reason given by the GPs involved in the Delphi process. Perhaps I am scarred by my own anecdotal experience of two primary school age children with Wilms tumour who presented with very advanced disease, several months after their parent recalled having seen a single episode of blood in urine and being reassured in primary care. Suggest this is an area that needs better evidence as a red flag warning signal for immediate referral.
--

REVIEWER	Reviewer name: Dr. Juliet Gray Institution and Country: 27 Locks Road, United Kingdom of Great Britain and Northern Ireland Competing interests: None
REVIEW RETURNED	29-Jan-2023

GENERAL COMMENTS	The previous reviewers' comments have been taken into account, and the manuscript edited accordingly.
---

REVIEWER	Reviewer name: Dr. Bob Phillips Institution and Country: University of York, United Kingdom of Great Britain and Northern Ireland Competing interests: None
REVIEW RETURNED	28-Jan-2023

GENERAL COMMENTS	Excellent modification with improvement in the introduction/justification and description of the process of moving from symptom-review to the major symptoms.
---

VERSION 2 – AUTHOR RESPONSE